

# A fifteen-million-year surface- and subsurface-integrated TEX₈₆ temperature record from the eastern equatorial Atlantic

Carolien M. H. van der Weijst[1], Koen J. van der Laan[1], Francien Peterse[1], Gert-Jan Reichart[1,2], Francesca Sangiorgi[1], Stefan Schouten[1,2], Tjerk J.T. Veenstra[1], Appy Sluijs[1]

[1]Department of Earth Sciences, Utrecht University, 3584 CB Utrecht, the Netherlands
[2]NIOZ Royal Netherlands Institute for Sea Research, 1797 SZ 't Horntje, the Netherlands

*Correspondence to*: C.M.H. van der Weijst (c.m.h.vanderweijst@uu.nl)

**Abstract.**

TEX$_{86}$ is a paleothermometer based on Thaumarcheotal glycerol dialkyl glycerol tetraether (GDGT) lipids and is one of the most frequently used proxies for sea-surface temperature (SST) in warmer-than-present climates. However, the calibration of TEX$_{86}$ to SST is controversial because its correlation to SST is not significantly stronger than that to depth-integrated surface

to subsurface temperatures. Because GDGTs are not exclusively produced in and exported from the surface ocean, sedimentary GDGTs may contain a depth-integrated signal that is sensitive to local subsurface temperature variability, which can only be proved in downcore studies. Here, we present a 15 Myr TEX$_{86}$ record from ODP Site 959 in the Gulf of Guinea and use additional proxies to elucidate the source of the recorded TEX$_{86}$ variability. Relatively high GDGT[2/3] ratio values from 13.6 Ma indicate that sedimentary GDGTs were partly sourced from deeper (>200 m) waters. Moreover, late Pliocene TEX$_{86}$

variability is highly sensitive to glacial-interglacial cyclicity, as is also recorded by benthic $\delta^{18}$O, while the variability within dinoflagellate assemblages and surface/thermocline temperature records ($U^{k'}_{37}$ and Mg/Ca), is not primarily explained by glacial-interglacial cyclicity. Combined, these observations are best explained by TEX$_{86}$ sensitivity to sub-thermocline temperature variability. We conclude that the TEX$_{86}$ record represents a depth-integrated signal that incorporates a SST and a deeper component, which is compatible the present-day depth distribution of Thaumarchaeota and with the GDGT[2/3]

distribution in core tops. The depth-integrated TEX$_{86}$ record can potentially be used to infer SST variability, because subsurface temperature variability is generally tightly linked to SST variability. Using a subsurface calibration with peak calibration weight between 100-350 m, we estimate that east equatorial Atlantic SST cooled by ~5°C between the Late Miocene and Pleistocene. On shorter timescales, we use the TEX$_{86}$ record as an Antarctic Intermediate Water (AAIW) proxy and evaluate climatological leads and lags around the Pliocene M2 glacial (~3.3 Ma). Our record, combined with published information,

suggests that the M2 glacial was marked by AAIW cooling during an austral summer insolation minimum, and that decreasing CO$_2$ levels were a feedback, not the initiator, of glacial expansion.



## 1. Introduction

Accurate tropical sea-surface temperature (SST) reconstructions are needed to assess oceanic heat distribution and ocean-atmosphere circulation in warmer-than-present climates, such as during the Pliocene. Tropical warming, even of a small magnitude, can lead to major changes in atmospheric circulation, with effects including intensification of monsoonal precipitation (Haywood et al., 2020; Zhang et al., 2016) and warming of the extratropics via teleconnections (Barreiro et al., 2006). At present, the most frequently used proxies for past SSTs are Mg/Ca (foraminifer calcite), $U^{k'}_{37}$ (alkenones) and TEX$_{86}$ (glycerol dialkyl glycerol tetraethers; GDGTs). Each proxy has specific confounding factors and calibration issues. For example, Mg/Ca paleothermometry requires corrections for the Mg/Ca ratio of seawater and diagenesis (Dekens et al., 2002; Evans et al., 2016), and the $U^{k'}_{37}$ proxy is insensitive to temperatures above ~28°C (Müller et al., 1998). TEX$_{86}$ is potentially one of the best proxies to reconstruct SSTs above 28°C, but one critical concern is that GDGTs are produced by marine Thaumarchaeota throughout the water column and often dominantly below the mixed layer (Ingalls et al., 2006; Kim et al., 2015; Lengger et al., 2014; Shah et al., 2008).

The export depth of downcore GDGTs can potentially be traced with the fractional abundance of GDGT-2 to GDGT-3 (GDGT[2/3] ratio; Hernández-Sánchez et al., 2014; Pearson et al., 2016; Taylor et al., 2013; Villanueva et al., 2015). GDGT[2/3] values are typically low (<5) in suspended particulate matter (SPM) sampled from shallow (<100 m) water, and sharply increase to >25 in deeper water (Hurley et al., 2018). This division is currently best explained by depth-differentiation of Thaumarcheotal ecotypes with different cyclization patterns (Kim et al., 2016; Villanueva et al., 2015). Core tops do not have GDGT[2/3] values indicative of a purely "deep" Thaumarcheotal GDGT source (Figure 1). Instead, they show a gradual transition from low to moderately high values, which can be attributed to the integration of GDGTs from a range of shallow to intermediate depths. Likewise, TEX$_{86}$ (Equation 1) may reflect a depth-integrated surface to subsurface signal (SubST), that is sensitive to temperature variability at and below the surface ocean (Kim et al., 2008, 2012; Schouten et al., 2002; Tierney and Tingley, 2015).

$$\text{TEX}_{86} = \frac{[\text{GDGT-2} + \text{GDGT-3} + \text{cren'}]}{[\text{GDGT-1} + \text{GDGT-2} + \text{GDGT-3} + \text{cren'}]} \qquad (1).$$

Indeed, several multi-proxy paleostudies suggest that dissimilarities between TEX$_{86}$-based SST reconstructions and other temperature records may be explained by a SubST signal recorded in TEX$_{86}$ variability (e.g. Lopes dos Santos et al., 2010; McClymont et al., 2012; Rommerskirchen et al., 2011; White and Ravelo, 2020). Together with the GDGT[2/3] distribution in core tops (Figure 1), these downcore records support the use of a SubST calibration. However, the targeted depth interval and weight distribution of the temperature integration remain subject of discussion (Ho and Laepple, 2016, 2017; Tierney et al., 2017). The correlation of core top TEX$_{86}$ to SST is not significantly stronger than to SubST, as obtained from a wide range of depth intervals (Ho and Laepple, 2016; Schouten et al., 2002). Therefore, the sensitivity of sedimentary TEX$_{86}$ to





temperature cannot be inferred from variability in the spatial dimension but variability in the temporal dimension may provide

solutions. Here, we present a new 0-15 Ma TEX$_{86}$ record from Ocean Drilling Program (ODP) Site 959 in the eastern equatorial

Atlantic. We evaluate the TEX$_{86}$ and GDGT[2/3] evolution on million-year and glacial-interglacial timescales during the late

Pliocene. We complement these records with U$^{k'}_{37}$ SST estimates and dinoflagellate cyst (dinocyst) assemblages and compare

them to the Site 959 benthic $\delta^{18}$O record (Norris, 1998a; van der Weijst et al., 2020) and Mg/Ca-based SST and thermocline

temperature records (van der Weijst et al., 2021) to elucidate the local source of downcore TEX$_{86}$-index variability and,

consequently, paleoclimatic change.

## 2 Material and Methods

### 2.1 Site, age model and sampling

ODP Site 959 was drilled during Leg 159 in the Gulf of Guinea, ~160 km offshore Ghana and Ivory Coast (3.62°N, 2.73°W;

2090 m depth; Mascle et al., 1996). It is presently located below the eastward flowing Guinea Current, which originates from

the North Equatorial Counter Current and the Canary Current (Norris, 1998a). SST currently varies between 25.3-28.6°C on a

seasonal time scale (Locarnini et al., 2013) and surface salinity ranges between 34.6-35.0 (Zweng et al., 2013). The water

column is characterized by a shallow thermocline, with the 20°C isotherm depth annually varying between ~40-60 m in

response to coastal upwelling (Locarnini et al., 2013, van der Weijst et al. 2021). A minor upwelling event occurs in boreal

winter and a longer and stronger event in boreal summer (Djakouré et al., 2017; Verstraete, 1992; Wiafe and Nyadjro, 2015).

We studied the 0-160 m interval of Hole C and 162-194 m interval of Hole A. The lithology gradually changes from

nannofossil/foraminifer chalk at the bottom to nannofossil/foraminifer ooze at the top (Mascle et al., 1996). The age model is

based on nannofossil biostratigraphy (Shin et al., 1998) and benthic $\delta^{18}$O stratigraphy (Wagner, 1998) between 0-23 mbsf,

benthic $\delta^{18}$O stratigraphy between 33-46 mbsf (van der Weijst et al., 2020), astronomical tuning of high resolution X-ray

fluorescence data (Vallé et al., 2016) between 51-100 mbsf, and a spline function through planktic foraminifer and nannofossil

biostratigraphy between 162-194 mbsf (Norris, 1998b; Shafik et al., 1998, Veenstra et al in prep.). We plot the data between

21.35-97.43 mbsf on the revised Meters Composite Depth (rMCD) scale of Vallé et al. (2016). Because outside this interval,

a splice is unavailable, we constructed a revised Meters Below Sea Floor (rMBSF) scale in coherence with Vallé et al. (2016)

and Veenstra et al. (in prep.).

The core was sampled on a relatively low resolution between 0-15 Ma for biomarker analysis and on a higher resolution in the

late Pliocene (2.7-3.5 Ma) interval for multi-proxy reconstructions. A total of 219 samples were extracted for TEX$_{86}$, of which

are closely spaced in the 2.7-3.5 Ma interval. U$^{k'}_{37}$ data was generated for 79 of these samples, of which 52 from the 2.7-

3.5 Ma interval. Additionally, we performed palynological assessments on 48 late Pliocene splits of samples that were also

used for biomarkers analysis.



## 2.2 Biomarker proxies

At Utrecht University, sediment samples were freeze dried and the outer surfaces were removed using a clean knife to prevent
contamination. The sediment was finely ground using a mortar and pestle. In the process, a split of coarsely ground material
was taken for palynology. Biomarkers were extracted at 100 °C and 7.6 x $10^6$ Pa using a dichloromethane (DCM):methanol
(9:1, *v/v*) solvent mixture in an accelerated solvent extractor (ASE 350, Dionex). The total lipid extract was separated over an
activated an $Al_2O_3$ column into an apolar, ketone and polar fraction using hexane:DCM (9:1, *v/v*), hexane:DCM (1:1, *v/v*) and
DCM:methanol (1:1, *v/v*) as eluents, respectively. For GDGT analysis, a $C_{46}$ GDGT (99 ng) standard was added to the polar
fraction before it was filtered over a 0.45 µm PTFE filter and dissolved in hexane:isopropanol 99:1 (*v/v*). Measurements were
performed on a Ultrahigh-Performance Liquid Chromatography-Mass Spectrometer (UHPLC-MS) using the method of
Hopmans et al., (2016). Samples with a branched to isoprenoid tetraether (BIT) index of >0.3 could be influenced by
terrestrially produced GDGTs (Hopmans et al., 2004; Weijers et al., 2006) and were excluded from further analysis (n=3). All
samples were within the reliable range of the Methane Index (MI; Zhang et al., 2011) and Ring Index (RI; Zhang et al., 2016),
suggesting no appreciable inputs from methanogenic archaea or other non-thermal factors.

The ketone fraction was dissolved in hexane and analysed on a Gas Chromatograph (GC) coupled to a flame ionisation detector
(GC-FID, Hewlett Packard 6890 series) equipped with a CP-Sil 5 fused silica capillary column (25 m x 0.32 mm; film thickness
0.12 um) and a 0.53 mm precolumn. Samples were injected on-column at 70°C with helium as a carrier gas and a flow rate of
2 ml/min. The oven program was as follows: 70°C for 1 min, then ramped to 130°C at 20°C/min, then to 320°C at 4C/min,
and then held isothermal for 10 mins. $U^{k'}_{37}$ values were calculated from the fractional abundances of the $C_{37:2}$ and $C_{37:3}$
alkenones following Prahl and Wakeham (1987) and calibrated to SST using the calibration of Müller et al. (1998).

## 2.3 Palynology

Coarsely crushed freeze-dried samples were spiked with a known amount of *Lycopodium clavatum* spores to quantify absolute
palynomorph (Stockmarr, 1972) and treated with 30% HCl (2x) and 40% cold HF (2x) to remove carbonates and silicates. The
residue was sieved over nylon mesh, and from the 10-250 µm fraction, the lighter organic fraction was separated from the
heavy mineral fraction by suspension. The remaining palynological residue was mounted on glass microscopic slides with
glycerine jelly. Samples were analysed using an optical microscope at under 400× magnification and approximately 300
dinoflagellate cysts (dinocysts) were counted per sample. Taxonomy follows that of Williams et al. (2017). *Capisocysta* was
only identified on the genus level, because known *Capisocysta* species are best identified by the number of antapical plates
(Head, 1998), which is complicated by the common dissociation of the hypocystal plates. We calculate the relative abundances
of a taxon based on the total dinocyst sum. Dinocysts with affinities for cold-temperate waters in the modern ocean





(Boessenkool et al., 2001; Zonneveld et al., 2013) were grouped as "cool species" and used for estimating surface water
temperature. This species are: *Ataxodinium choane, Corrudinium devernaliae, Corrudinium harlandii, Impagidinium
pallidum, Nematosphaeropsis labyrinthus* and *Pyxidinopsis reticulata*. We also use the P/(P+G) ratio (Versteegh, 1994) as a
paleoproductivity index. The P/(P+G) ratio quantifies the number of cysts produced by heterotrophic dinoflagellates, which
are characterized by peridinioid type tabulation (P) over the total number of dinocysts, including those produced by autotrophic
or mixotrophic species with a gonyaulacoid type tabulation (G).

## 3 Results

### 3.1 TEX$_{86}$, U$^{k'}_{37}$ and GDGT[2/3] evolution at Site 959

Between 15 and 11 Ma, TEX$_{86}^{H}$-SST (Kim et al., 2010) fluctuates around an average value of ~29°C (Figure 2), similar to the
present-day local annual maximum surface temperature of 28.6°C. A long-term gradual cooling starts around 11 Ma and is
punctuated by two major cooling steps of ~2°C each, around 4.9 Ma and 3.4 Ma, in which TEX$_{86}^{H}$-SSTs drop well below the
present-day local annual minimum surface temperature of 25.3°C. The cooling trend seems to end in the late Pleistocene,
where TEX$_{86}^{H}$-SST records a ~4°C warming between 0.5 Ma and present, with a core top TEX$_{86}^{H}$-SST of 24.0°C. U$^{k'}_{37}$ is at
saturation (~28°C) for most of the interval but registers some colder temperatures in the late Pliocene (27-28°C), and drops
below 27°C after 1.8 Ma. The core top sample registers a temperature of 26.6°C, which is within calibration error (1.5 °C;
Müller et al., 1998) of the 27.5°C modern mean annual SST (Locarnini et al., 2013). The GDGT[2/3] ratio is 7.0 in the core
top. In the oldest part of the studied interval, GDGT[2/3] values vary between 4 and 5.5. Following an abrupt increase at ~13.6
Ma, GDGT[2/3] varies largely between 5 and 8. After 1.7 Ma, GDGT[2/3] persistently increases to >7.

### 3.2 Pliocene dinocyst assemblages

Palynological assemblages are dominated by dinocysts with only minor amounts of terrestrially derived pollen and spores.
Dinocyst preservation is generally good. The assemblages are dominated by *Brigantedinium* spp. (19-60%), closely followed
by *Spiniferites* spp. (8-36%) (Figure 3; see supplementary information for a complete overview of the assemblages). The
interval between 3.33 Ma and 3.16 Ma was sampled at a higher resolution (~5 kyr on average) and displays strong variability
of the negatively correlated *Brigantedinium* spp. and *Spiniferites* spp. on timescales <10 kyr. Other relatively abundant and
consistently present taxa are *Operculodinium* spp., *Pentapharsodinium dalei* and *Impagidinium* spp. Peak abundances of
*Lingulodinium* spp. were recorded at ~3.26 Ma (*Lingulodinium machaerophorum*) and at 2.77 Ma (*Lingulodinium
hemicystum*). *Capisocysta* sp. is consistently present in the lower part of the interval, with peak abundances (>20%) at 3.32
Ma and 3.20 Ma but has its last occurrence at 3.17 Ma. There is general coherence between TEX$_{86}^{H}$-SST variability and the
relative abundance of the "cool species" group, which is dominated by *N. labyrinthus*. On average, the dinocysts with affinities
for colder-temperate waters make up 6% of the total assemblage, but peak abundances of 10-13% occur during the M2 glacial.
The P/(P+G) paleoproductivity index is 0.48 on average, with minimum and maximum values of 0.23 and 0.70 respectively





(Figure 3). It is primarily driven by the abundance of *Brigantedinium* spp. The P/(P+G) ratio shows no significant relationship with $TEX_{86}^H$-SST ($R^2$=0.05, p=0.14 in linear regression analysis).

## 4. Discussion

### 4.1 Identifying the source depth of the $TEX_{86}$ signal

The 15-million-year $TEX_{86}^H$-SST record at Site 959 shows a cooling from >30°C between 11-15 Ma to <20°C between 0-3.3
Ma (Figure 2). The $U^{k'}_{37}$ record, on the other hand, is near or at proxy saturation (>27°C) for most of the interval, with some lower values after 1.8 Ma. Assuming that both proxies reflect SST, the relatively small offset between $U^{k'}_{37}$ and $TEX_{86}^H$-SST offset in the core top sample (2.7°C) could be explained by a combination of calibration errors and seasonality. However, the average downcore offset between $TEX_{86}^H$-SST and $U^{k'}_{37}$ is 5.4°C and peaks at 7-8°C during the late Pliocene interglacials (Figure 4 and Figure 5), which demonstrates that $TEX_{86}^H$-SST underestimates past SST at this site, assuming that $U^{k'}_{37}$ is a
more accurate representation of SST. This could potentially be resolved by using an alternative, regional SST calibration. The BAYSPAR-SST calibration (Tierney and Tingley, 2014) accounts for spatial variability of the $TEX_{86}$-SST relationship and is the only available calibration that raises absolute SST estimates, but like the $TEX_{86}^H$-SST (Kim et al., 2010) and low-latitude Atlantic (Zhang et al., 2018) calibrations, it likely overestimates SST variability (Figure S1), indicating that the calibration slope is too steep. This is potentially a problem inherent to the calibration of the $TEX_{86}^H$-index to SST, and could reflect
downcore $TEX_{86}$ sensitivity to SubST instead of SST variability (Ho and Laepple, 2016). In the following sections, we explore multi-proxy data from Site 959 in search of indications for the water depth the downcore $TEX_{86}$ signal is reflecting.

### 4.1.1 GDGT[2/3] values

A substantial contribution of GDGTs from below the oceanic surface layer is supported by the relatively high GDGT[2/3]
ratio, which fluctuates between 5 and 8 in the majority of the record (Figure 2), values that are rarely observed in modern day shallow water (<100 m) SPM and surface sediments (e.g. Besseling et al., 2019; Hernández-Sánchez et al., 2014; Hurley et al., 2018). These values are best explained as a mixed signal of GDGTs from the upper 200 m and intermediate waters (Figure 1c), i.e., likely dominantly from below the thermocline (~50 m) at Site 959. A cross-plot of $TEX_{86}^H$-SST and GDGT[2/3] (Figure 6) shows that most data are clustered around a positive regression slope, but the oldest (>13.6 Ma) and youngest (<1.7
Ma) data plot outside this main cluster, potentially signaling systematic changes in GDGT production and export depth. Both 13.6 Ma and 1.7 Ma mark the onset of a strong GDGT[2/3] increase (Figure 2), signaling systematic deepening of the source of the sedimentary GDGT signal.

The ratio of GDGT[2] to GDGT[3] does not directly affect the $TEX_{86}$-index (Equation 1). If they both covaried in response to
GDGT export depth, then samples with a higher GDGT[2/3] ratio should be associated with lower $TEX_{86}^H$-SST values. The





positive relationship between TEX$_{86}$ and GDGT[2/3] (Figure 6) suggests that the long term TEX$_{86}$ trends are not driven by GDGT export depth, but reflect overall temperature changes instead. Alternatively, non-thermal factors such as water column oxygenation and nutrient supply may have influenced GDGT cyclization: high nutrient availability and ammonia oxidation rates have been linked to low TEX$_{86}$ values (Hurley et al., 2016; Park et al., 2018). However, following the GDGT[2/3] shift

at 13.6 Ma, TEX$_{86}$$^H$-SST does not systematically change until ~11 Ma (Figure 2). Moreover, late Pliocene dinocyst assemblages (Figure 3) are characterized by a highly variable P/(P+G) ratio, indicating that upwelling and nutrient supply were highly variable on sub-Milankovitch timescales. In contrast to TEX$_{86}$$^H$-SST (Figure 4), the P/(P+G) ratio shows no glacial-interglacial variability. It is therefore unlikely that TEX$_{86}$ variability was primarily driven by non-thermal factors.

**4.1.2 Surface, thermocline and sub-thermocline temperature variability**

The late Pliocene TEX$_{86}$$^H$-SST record follows a different evolution than the other SST proxies (Figure 4). Whereas TEX$_{86}$$^H$-SST decreases by ~2°C, U$^{k'}_{37}$ and Mg/Ca-based (van der Weijst et al., 2021) SSTs show no significant net change in this interval. Moreover, although Mg/Ca and U$^{k'}_{37}$ register some cooling during the late Pliocene glacials, their variability seems not primarily driven by glacial-interglacial cyclicity, as is best observed in the high-resolution interval from M2 to KM2

(Figure 4). This is also true for the relative abundance of dinocysts with affinities for colder temperatures (Figure 3 Figure 4). Furthermore, around the KM2 and M2 glacials, TEX$_{86}$$^H$-SST cooling leads increasing abundances of cool dinocysts, as well as Mg/Ca$_{(G.ruber)}$ and U$^{k'}_{37}$ cooling by ~5-10 ka (Figure 5), which underscores the independent evolution of these records.

At sites where TEX$_{86}$ is suspected to be affected by GDGTs produced below the surface ocean, TEX$_{86}$$^H$-SST has occasionally

been interpreted to reflect thermocline temperature variability (e.g. Lopes dos Santos et al., 2010). Thermocline temperatures are lower and can be highly variable through time, which potentially explains both absolute SSTs underestimation and amplitude overestimation in specific TEX$_{86}$$^H$-SST records. However, both could also be explained as a general artefact of an overestimated calibration slope that results from calibrating a depth-integrated GDGT signal to SST (Ho and Laepple, 2015, 2016). Our TEX$_{86}$$^H$-SST record shows weak coherence with the Site 959 Mg/Ca$_{(N.dutertrei)}$ thermocline temperature record

(Figure 4; van der Weijst et al., 2021). Whereas TEX$_{86}$ registers late Pliocene cooling, Mg/Ca$_{(N. dutertrei)}$ registers warming in relation to thermocline deepening (van der Weijst et al., 2021). Moreover, while the range of Mg/Ca$_{(N. dutertrei)}$ temperatures is similar to TEX$_{86}$$^H$-SSTs on glacial-interglacial timescales, Mg/Ca$_{(N. dutertrei)}$ does not follow the glacial-interglacial cyclicity as recorded in the TEX$_{86}$.

In contrast, the variability in the TEX$_{86}$$^H$-SST and benthic $\delta^{18}$O records at Site 959 is very similar (Figure 2, Figure 4). Benthic $\delta^{18}$O is a faithful recorder of glacial-interglacial cyclicity because it registers a combined signal of deep ocean temperature and polar ice sheet volume. Could TEX$_{86}$ at Site 959 also be sensitive to glacial-interglacial cyclicity in deeper waters? At the onset of the M2 and KM2 glacials, the glacial expression of TEX$_{86}$$^H$-SST leads $\delta^{18}$O by ~5 kyr (Figure 5), which indicates that,



despite the similarities between $TEX_{86}^{H}$-SST and benthic $\delta^{18}O$, these proxies record variability in different water masses.

Bottom waters at Site 959 are predominantly ventilated by North Atlantic Deep Water (NADW), whereas depths below the thermocline are occupied by South Atlantic Central Water (SACW; Figure 7). SACW is a mix of surface/thermocline waters and Antarctic Intermediate Water (AAIW), with an increasing proportion of AAIW with depth. It is formed at the Antarctic Polar Front, and is therefore, like benthic $\delta^{18}O$, sensitive to high-latitude climate change. AAIW dynamics have previously been suggested to be the main driver of $TEX_{86}$ variability in the Arabian Sea (Huguet et al., 2006) and in the southeast Atlantic

(Rommerskirchen et al., 2011). If $TEX_{86}$ at Site 959 records a mixed signal from surface and subsurface waters, as is supported by the relatively high GDGT[2/3] ratio (Figure 1 and Figure 2), it is likely sensitive to AAIW dynamics, which explains the strong coherence with the benthic $\delta^{18}O$ record. Collectively, these observations suggests that the downcore $TEX_{86}$ record at Site 959 is substantially affected by temperature variability below the surface ocean, and that calibration to SubST is more appropriate than to SST.


## 4.2 Calibration of the depth-integrated $TEX_{86}$ record

Several $TEX_{86}$-SubST calibrations are available, with each assuming, among other things, a different depth-integration of the water column (Figure 7). The BAYSPAR-SubST (Tierney and Tingley, 2014) and $TEX_{86}^{H}$-SubST (Kim et al., 2012) calibrations both target the upper 200 m of the water column, but the weight of the BAYSPAR-SubST calibration is centered

at relatively shallow depths compared to the linearly weighted $TEX_{86}^{H}$-SubST calibration (Figure 7). The HL16-SubST (Ho and Laepple, 2016) calibration targets the upper 950 m. The validity of this depth interval was questioned (Tierney et al., 2017), but with peak calibration weight at 100-350 m (Figure 7), it is compatible with the documented vertical distribution of Thaumarcheotal cells counts and GDGT concentrations (e.g. Hernández-Sánchez et al., 2014; Schouten et al., 2012; Wuchter et al., 2005). Moreover, based on the large proportion of core tops with higher (>5) GDGT[2/3] values in the global calibration

set, a slightly deeper calibration target (including depths below 200 m) is arguably preferable. This deeper calibration target lowers the slope of the calibration, which dampens reconstructed Miocene-Pliocene temperature variability at Site 959 (Fig. S1).

The ratio between temperature change in the surface and subsurface ocean is 1:1 when averaged across many sites and on

longer timescales (Ho and Laepple, 2016). Depending on the calibration, the magnitude of Late Miocene to Pleistocene cooling at Site 959 is 10°C (BAYSPAR-SubST), 7°C ($TEX_{86}^{H}$-SubST) or 5°C (HL16-SubST; Figure 7). Assuming a 1:1 ratio, the long-term SST trend should be of a similar magnitude. Currently available tropical SST estimates on the studied timescale are either compromised by saturation of the $U_{37}^{k'}$ proxy, or are based on $TEX_{86}$, and are therefore not suitable for independent comparison. However, global mean surface temperature estimates based on benthic $\delta^{18}O$ data suggest a ~5°C cooling across

the same interval (Hansen et al., 2013; Tierney et al., 2020). In the absence of major oceanographic changes, it is unlikely that the local cooling trend at Site 959 was larger, because temperature variability is generally lower in the tropics compared to





high latitudes where deep-ocean waters derive from (i.e., polar amplification). This suggests that the BAYSPAR-SubST and TEX$_{86}^{H}$-SubST calibrations may overestimate the magnitude of long-term cooling (Figure 7). From the currently available calibrations, HL16-SubST best approximates both expected glacial-interglacial variability and multi-million year trends

(Figure 7 and Figure S1), and could potentially be used in combination with other SST records and/or modern water column data to correct for the local SST-SubST offset (Figure S1). It is possible that the 1:1 ratio between SST and SubST did not persist across the entire record, e.g., due to changes in AAIW production (Holbourn et al., 2013). To further improve tropical SST estimates in warmer climates, it should be further explored under which conditions a TEX$_{86}$ record that is driven by subsurface variability can be used to reconstruct SST variability, and how to correct for the surface-subsurface temperature

offset.

### 4.3 Late Neogene TEX$_{86}$ as an intermediate ocean signal at Site 959: New insights in M2 glacial inception

It is currently unclear what caused M2 glacial inception, but hypothesized mechanisms include declining atmospheric CO$_2$ levels (Berends et al., 2019; Dolan et al., 2015), reduced latitudinal heat transport by the North Atlantic Current (de Schepper

et al., 2009; 2013) on the Northern Hemisphere (NH) and by Indonesian throughflow (De Vleeschouwer et al., 2018) on the Southern Hemisphere (SH). Multi-proxy data from Site 999 in the Caribbean Sea show that declining CO$_2$ levels lagged the benthic δ$^{18}$O glacial expression at the onset of M2 (de la Vega et al., 2020). This indicates that declining CO$_2$ levels did not initiate the M2 glacial, although they might have affected its intensity and duration.

Because the TEX$_{86}$ record at Site 959 is affected by AAIW, it can potentially be used to study the connection between high and low latitude climate change. The TEX$_{86}$ and benthic δ$^{18}$O records at Site 959 correspond closely but are out-of-sync during the M2 glacial (Figure 5). Both records display a double peak around M2, but TEX$_{86}$ leads δ$^{18}$O by ~5 kyr. Whereas the depth-integrated TEX$_{86}$ signal at Site 959 is sensitive to SH conditions through AAIW, bottom waters at Site 959 consisted predominantly of NADW sourced from the NH during the late Pliocene (van der Weijst et al., 2020). The TEX$_{86}$ lead therefore

indicates a lead of SH over NH cooling/glaciation at the onset of M2, in agreement with transient ice sheet simulation (Berends et al., 2019).

We further explore the relative chronology around M2 in Figure 8. The age models were aligned on peak glacial δ$^{18}$O values and the same records are plotted on the original age models in Figure S2. Although it is difficult to discriminate between signal

and noise at this resolution, some patterns seem to emerge from this compilation. Directly before M2, benthic δ$^{18}$O, SubST and SST records from the eastern equatorial Atlantic (Site 959 and Site 662) and Caribbean Sea (Site 999) indicate gradual, widespread cooling, while CO$_2$ levels were rising. A sharp drop in TEX$_{86}^{H}$-SubST indicates intermediate ocean cooling at the onset of M2, leading pronounced glacial cooling (benthic δ$^{18}$O) of the deep ocean and tropical surface ocean (best observed at Site 662). According to the chronology in Figure 8, this intermediate ocean cooling occurred during a SH summer insolation



minimum, suggesting that astronomical forcing may have amplified Southern Ocean cooling and tipped the system to glacial
conditions. The sensitivity of the Antarctic ice sheet to astronomical forcing is also reflected by the 3.3 Ma onset of ~20 kyr
cyclicity in Ice Rafted Debris (IRD) at East Antarctic Site IODP Site U1361, with peak IRD mass accumulation rates at SH
summer insolation minima (Patterson et al., 2014). Based on the chronology of Figure 8, SH cooling led atmospheric $CO_2$
drawdown by ~15-20 kyr. Lags of a similar magnitude were not unusual during Pleistocene glacial inceptions (Petit et al.,

1999). Marginal Antarctic diatom assemblages and lithology suggest that considerable sea ice expansion occurred during M2,
which could have inhibited $CO_2$ outgassing from the deep ocean (Ishino and Suto, 2020; McKay et al., 2012). Moreover, a
slight increase in dust-mediated iron fertilization of the Southern Ocean may have facilitated export productivity (Martínez-
Garcia et al., 2011). In combination with reduced mixing between Northern Component Water and Southern Component Water
(van der Weijst et al., 2020), these processes would have led to increased carbon storage in the deep ocean in response to M2

cooling.

## 5. Conclusions

Several lines of evidence show that the $TEX_{86}$ record at Site 959 in the eastern equatorial Atlantic is best explained as a depth-
integrated signal that is substantially affected by temperature variability below the thermocline. Relatively high GDGT[2/3]
ratios since ~13.6 Ma indicate that the sedimentary GDGTs were partly sourced from deeper (>200 m) waters. Moreover, late

Pliocene multi-proxy data shows that $TEX_{86}$ variability is predominantly driven by glacial-interglacial variability, in contrast
to other SST and thermocline records ($U^{k'}_{37}$ and $Mg/Ca_{(G.ruber)}$ SST and $Mg/Ca_{(N.duterrei)}$) and dinocyst abundances. The $TEX_{86}$
record strongly resembles the benthic $\delta^{18}O$ record, indicating that it dominantly records temperature changes in a sub-
thermocline water mass. At Site 959, intermediate depths are occupied by SACW, a mix between surface and AAIW.
According to the present-day water column composition, a substantial contribution of GDGTs from deeper (>200) waters is

needed to obtain $TEX_{86}$ sensitivity to AAIW. This favors a subsurface calibration that integrates across a wider range of water
depths, such as HL16-SubST (Ho & Laepple, 2016), in which calibration weight peaks 100-350 m. This calibration target
interval is compatible with pelagic Thaumarcheotal cell counts and GDGT concentrations, and with core top GDGT[2/3]
values in the global calibration set.

Even if $TEX_{86}$ mainly relects SubST variability, it may be a used to reconstuct past SSTs if the temporal relationship between
SST-SubST in a certain region is well-understood. Assuming a 1:1 relationship between long-term SubST and SST trends, our
$TEX_{86}$ record suggests 5°C tropical SST cooling between the Late Miocene and Pleistocene when calibrated with HL16-
SubST. Additionally, $TEX_{86}$ is also highly informative as a SubST proxy, because the sensitivity of $TEX_{86}$ to AAIW at Site
959 offers a chance to explore connections between high and low latitude climate variability. The $TEX_{86}$ and benthic $\delta^{18}O$

records at Site 959 are highly similar, but $TEX_{86}$ is sensitive to high latitude SH conditions through AAIW, whereas the benthic
$\delta^{18}O$ record is primarily forced by high latitude NH conditions through NADW. A ~5 kyr lead of $TEX_{86}$ relative to $\delta^{18}O$ at the



onset of the M2 glacial stage indicates a lead of SH over NH cooling. Multi-proxy data from Site 959, Site 999 (Caribbean Sea) and Site 662 (eastern equatorial Atlantic) aligned based on LR04 peak glacial $\delta^{18}O$ values indicate that AAIW/SH cooling also led tropical ocean cooling and $CO_2$ levels by 15-20 kyr, suggesting that $CO_2$ drawdown was a consequence of glacial

conditions. Instead, glacial expansion and AAIW cooling at SH insolation minima suggest that orbital forcing played a pivotal role in M2 glacial inception.

## Data availability

New Site 959 data are available as a supplement to this paper and will be uploaded to the PANGAEA online data repository upon publication.

**Competing interests**

The authors declare that they have no conflict of interest.

## Acknowledgements

We thank the International Ocean Discovery Program and the predecessor drilling programs for samples and data. We thank Arnold van Dijk, Giovanni Dammers, Natasja Welters, Klaas van Nierop and Dominika Kasjaniuk (Utrecht University) and

Wim Boer (NIOZ) for analytical support. This work was carried out under the program of the Netherlands Earth System Science Centre (NESSC), financially supported by the Ministry of Education, Culture and Science (OCW). AS thanks the European Research Council for Consolidator Grant 771497.

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





**Figure 1. (a): Depth distribution of GDGT[2/3] ratio in suspended particulate matter (SPM; solid lines) at several stations in the Atlantic Ocean (Hurley et al., 2018) and in core tops (circles) between 65°N-65°S (Tierney and Tingley, 2015). (b): GDGT[2/3] vs. TEX$_{86}$, with logarithmic fits for GDGT[2/3]>5 (blue line) an GDGT[2/3]<5 (orange line). (c): Depth-integrated (linear) GDGT[2/3] from SPM data (station 23; Hurley et al., 2018). Integration start depths at 0 m, 50 m, 100 m and 200 m, and continuous end depth on x-axis. Range of typical Site 959 sedimentary GDGT[2/3] values between orange dashed lines, and corresponding range of potential integration end depths in vertical. Note that the assumed linear depth-weighing may not be realistic, but that an integration start depth at 200 m is incompatible with the sedimentary GDGT[2/3] values at Site 959, regardless of depth-weighing.**

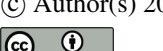



**Figure 2. New TEX$_{86}^{H}$-SST (Kim et al., 2010 calibration), U$^{k'}_{37}$-SST (Müller et al., 1998 calibration) and GDGT[2/3] records at Site 959, compared to benthic δ$^{18}$O records from Norris (1998a) and van der Weijst et al. (2020). Shaded interval and arrow indicate annual SST range and mean annual SST in the Gulf of Guinea, respectively (Locarnini et al., 2013).**



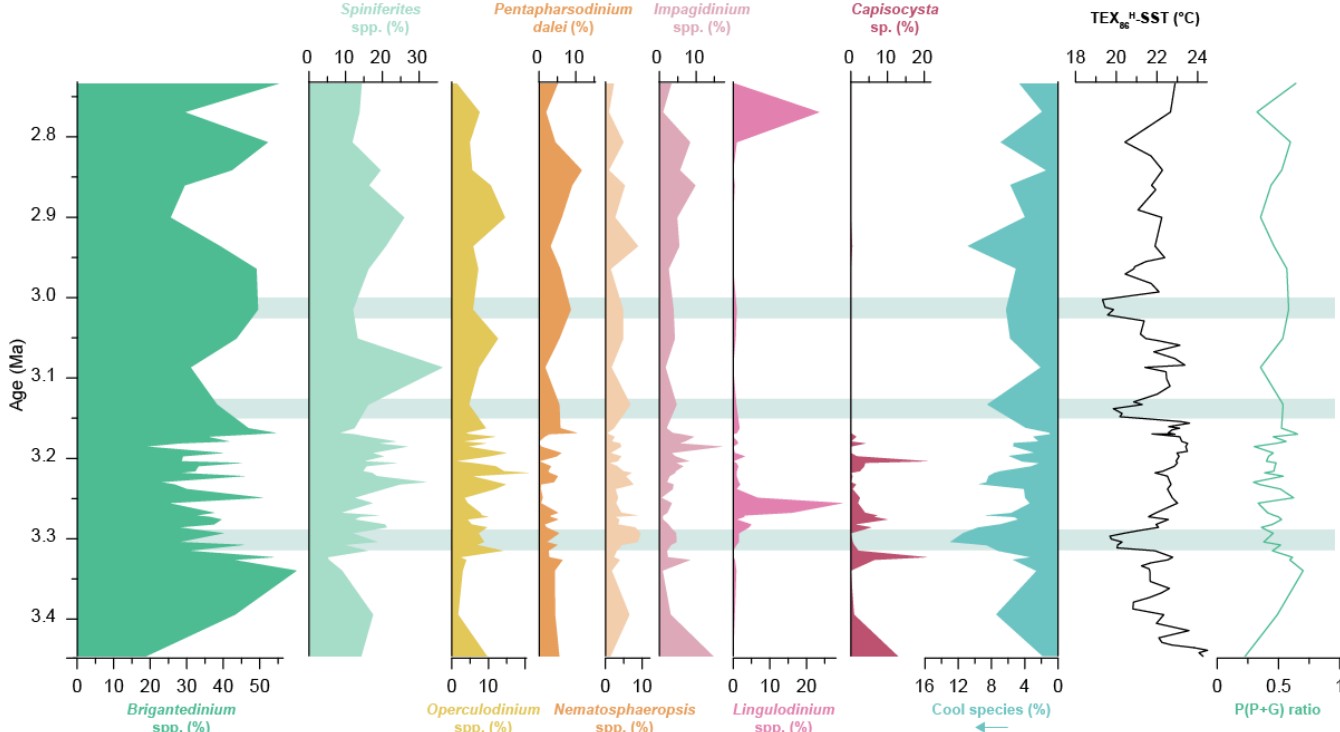

**Figure 3. Relative abundances of the major groups of dinoflagellate cysts, the sum of dinocysts with cold affinities ("Cool species"), and P/(P+G) paleoproductivity index compared to TEX$_{86}^{H}$-SST. Shaded bands indicate TEX$_{86}^{H}$-SST minima during the late Pliocene M2, KM2 and G20 glacials.**






**Figure 4. Compilation of late Pliocene multi-proxy temperature data and benthic δ¹⁸O at Site 959.** Mg/Ca$_{(G.ruber)}$ (pink) and
Mg/Ca$_{(N.dutertrei)}$ (yellow) from van der Weijst et al. (2021), benthic δ¹⁸O from van der Weijst et al. (2020) and LR04 δ¹⁸O stack
(Lisiecki and Raymo, 2005) for reference. Dashed lines in are LOESS smoothed trends and vertical bands indicate the M2, KM2
and G20 glacial stages.





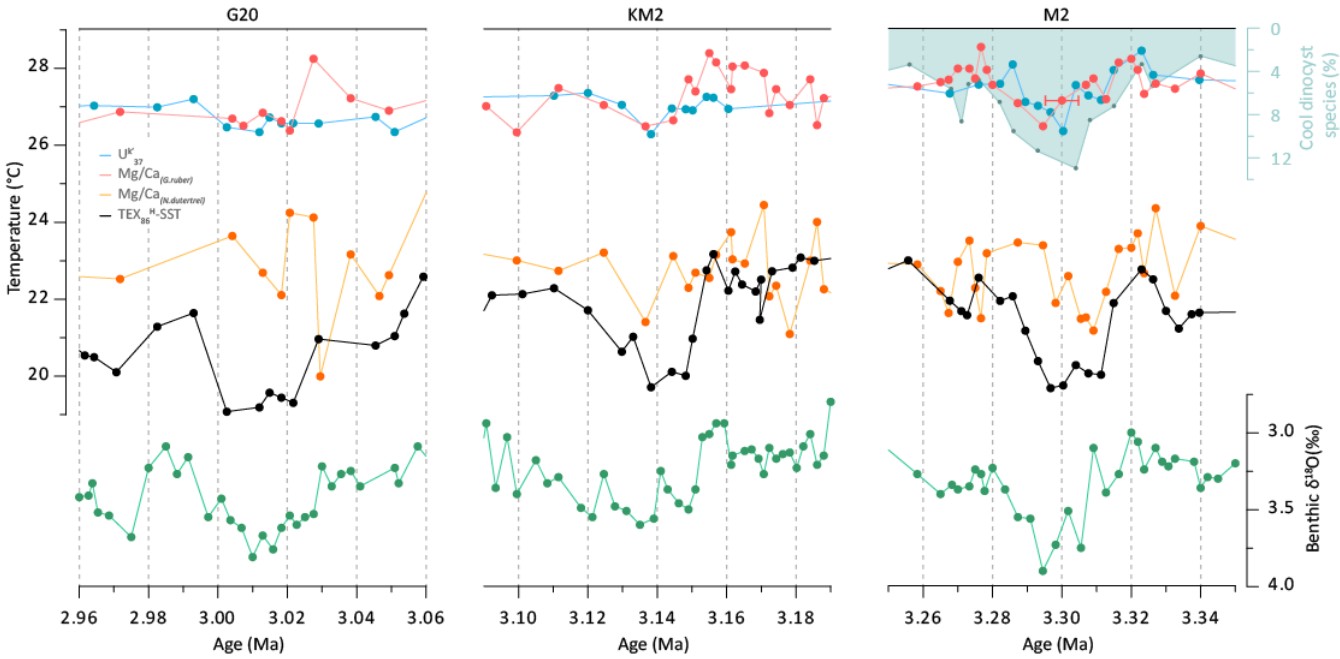

**Figure 5. Close-up of G20, KM2 and M2 glacial stages in 100 kyr windows. Multi-proxy temperature data and benthic δ¹⁸O as in Figure 4. The resolution of the dinocyst record was too low for meaningful comparisons in KM2 and M2 glacials.**





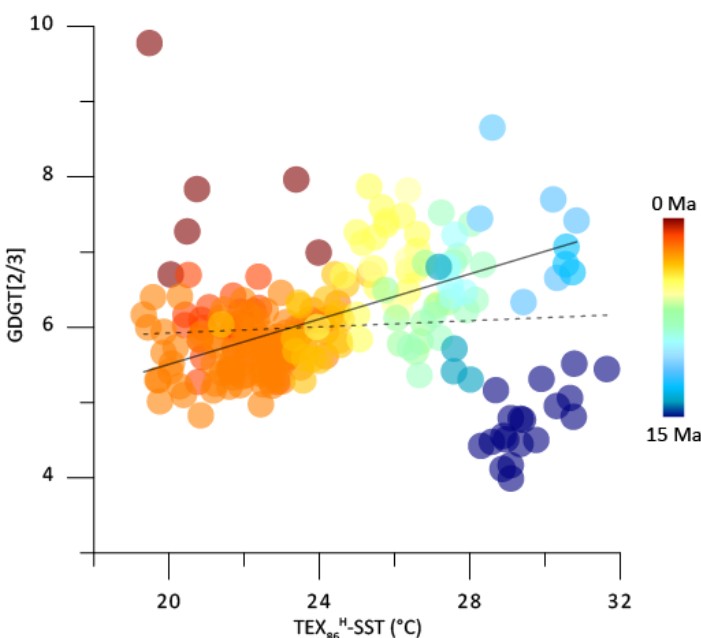

**Figure 6. Cross-plot of TEX$_{86}^{H}$-SST and GDGT[2/3]. The oldest (>13.6 Ma; dark blue) and youngest (<1.7 Ma; dark red) samples largely stand out from the majority of the data. Linear regression lines determined from full dataset (stippled line) and samples dated between 13.6 and 1.7 Ma (solid line; 88% samples, R$^2$=0.37, p<0.001).**



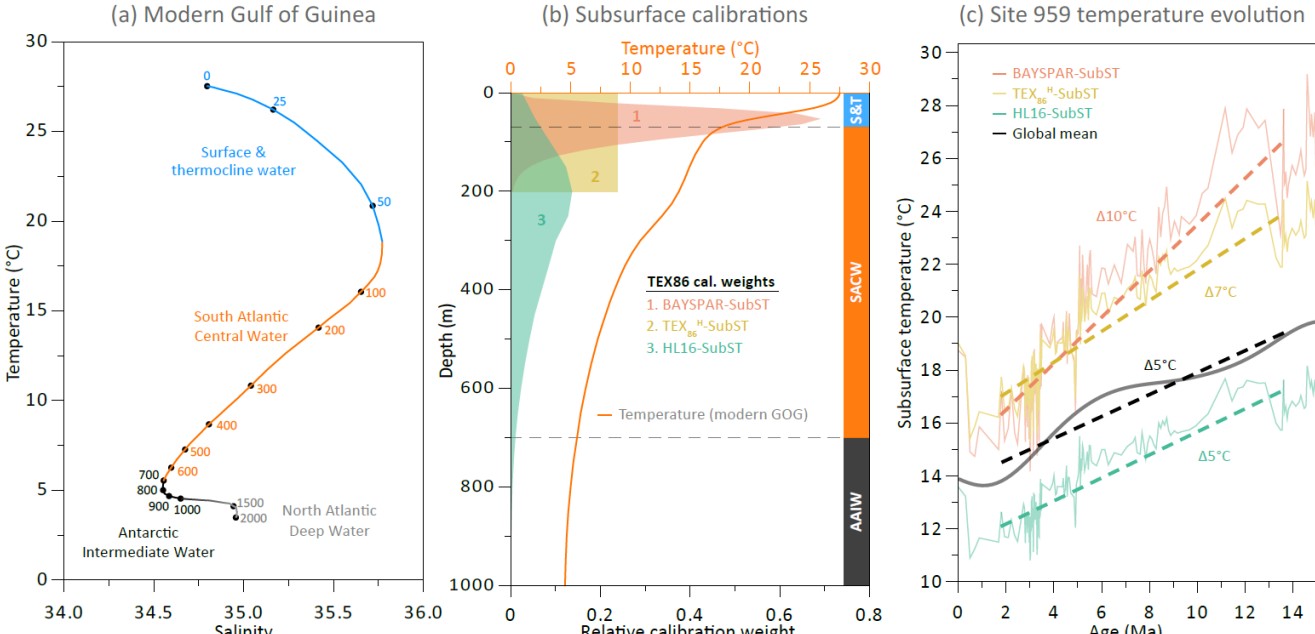

**Figure 7. (a): Temperature and salinity distribution along a depth transect in the modern Gulf of Guinea (numbers: meters below sea surface), data from Locarnini et al., (2013) and Zweng et al., (2013). (b): Vertical weight distribution of the discussed SubST calibrations (silhouettes) and the water column composition in the modern Gulf of Guinea. (c) Site 959 TEX86^H-SubST evolution according to different SubST calibrations, compared to global mean temperature evolution (Tierney et al., 2020). ΔTemperature estimated from linear trends (dashed lines) between the Middle Miocene and Pleistocene.**






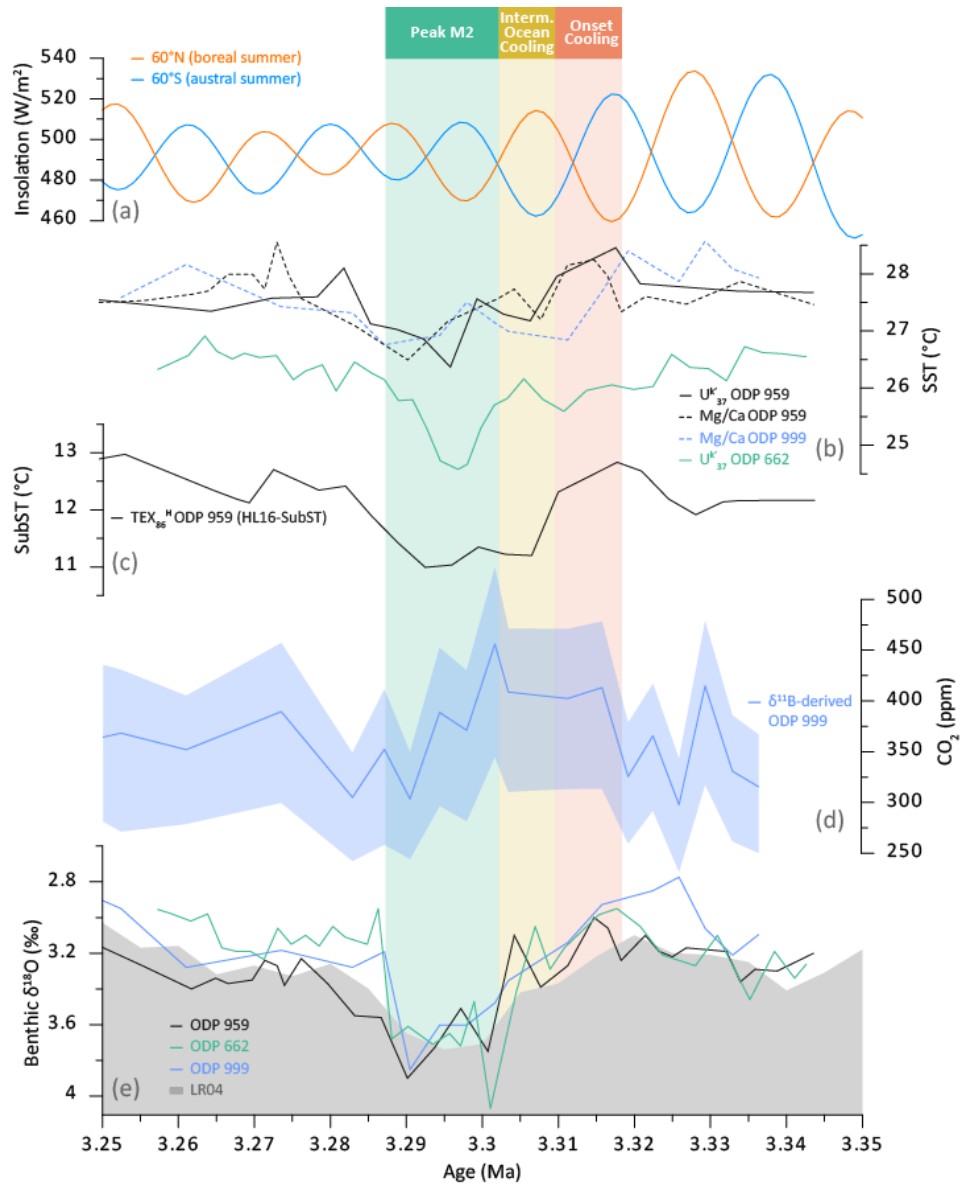

**Figure 8. Exploring the chronology around the M2 glacial with records aligned on peak glacial benthic $\delta^{18}O$ values (original age models in Figure S2). (a): Daily insolation at 60°N/S during NH/SH summer solstice (Laskar, 1990). (b): East equatorial Atlantic SST at Site 959 (this study) and Site 662 (Herbert et al., 2010) and Caribbean Sea SST at Site 999 (de la Vega et al., 2020). (c): Site 959 TEX$_{86}$ HL16-SubST, indicative of intermediate ocean temperatures (d): Atmospheric $CO_2$ reconstructions from Site 999 (de la Vega et al., 2020). (e): Benthic $\delta^{18}O$ records of Site 959 (van der Weijst et al., 2020), 662 (Lisiecki and Raymo, 2005) and 999 (Haug et al., 2001; de la Vega et al., 2020), aligned to LR04 (Lisiecki and Raymo, 2005) on peak M2 values.**