# Peer review of "A fifteen-million-year surface- and subsurface-integrated TEX86 temperature record from the eastern equatorial Atlantic"

_Climate of the Past, 2021_

## Author Comment (AC1)

Dear Editor,

We thank reviewer #1 for their comments and suggestions on our paper. Below, we discuss these point-by-point and indicate how we intend to adapt the manuscript in a revised version.

Sincerely, also on behalf of my co-authors,

Carolien van de Weijst

*Summary*

*The study by van der Weijst et al presents multiproxy-based temperature and productivity records spanning the past 15 Ma. Data are generated from an ODP core recovered from the eastern equatorial Atlantic, at a site that is under the influence of monsoon-induced upwelling. The proxies used are alkenone-based UK'37, archaeal GDGT-based TEX86 and dinocysts. Based on several lines of evidence, the authors argue that the TEX86 proxy records a mixture of surface and subsurface signal, likely that of Antarctic Intermediate Water (AAIW) as also suggested by two previous studies. Assuming that UK'37 reflects SST variability, benthic d18O records NADW variability and TEX86 reflects AAIW variability, in combination with other published temperature records and a pCO$_2$ record, the authors discuss possible mechanisms that lead to M2 glaciation.*

*General comments*

*Overall I find the manuscript very clear and well-written. The topic also fits well within the remit of the journal, thus will likely be of interest to the broad readership of Climate of the Past. Although generally accessible to the reader, I think some arguments can be further improved and/or need further clarification. I hope the authors will find my comments and suggestions helpful in revising the manuscript. Altogether it should amount to moderate revision. Below I outline my major concerns.*

REPLY: We thank reviewer #1 for their compliments regarding our work.

*(1) TEX86 reflects AAIW variability*

*The authors provided several lines of evidence to support their claim that TEX86 reflects subsurface temperature: high GDGT 2/3 ratio, TEX86H SSTs that are out of the modern range and unrealistically large magnitude of change over the last 15 Ma, temporal trends that are more similar to those of benthic d18O than UK'37 and Mg/Ca based on both mixed layer- and thermocline-dwelling species. The discussion is generally convincing.*

*What is however unclear to me is how the authors then link the TEX86 signal to AAIW variability. I find the explanation provided at Line 226-227 to be rather vague. The core-top value of their SubT TEX86 record is ~14ºC, which is a lot warmer than the temperature in AAIW (~5ºC according to Figure 7a) and is in fact closer to that of South Atlantic Central Water. Further, the core of depth range integrated by the HL16 calibration is 100-350 m, which is much shallower than the average depth of AAIW. As the climate interpretation hinges on this claim, the authors need to provide clearer arguments to support their statement. I wonder if it is possible to attribute the relative influence of SACW vs. AAIW using the depth distribution in the calibration used?*

REPLY: South Atlantic Central Water (SACW) present in the water column at Site 959 is formed in the South Atlantic subtropical gyre. Together with the Antarctic Intermediate water (AAIW) below (characterized by relatively low salinity and an increasing component with depth; see fig 7a) this forms the permanent thermocline waters. We agree with this reviewer that our signal represents a mix including both SACW and AAIW which hence must partly track AAIW variability. In other words, our TEX86 signal tracks Southern Ocean surface temperature variability imprinted in both SACW and AAIW. But clearly, we are sampling SACW so to comply with the comment of the reviewer, we will change reference of AAIW in the remainder of the manuscript to SACW, thereby noting that both represent a Southern Ocean climate signal.

*(2) Subsurface export of GDGTs*

*Most sedimentary GDGT 2/3 ratios along the core are > 5. The authors argue that this indicates that GDGTs in the sediment core are partially sourced from the subsurface ocean. I acknowledge that the GDGT 2/3 ratio is a routinely used indicator to cull TEX86 data, it does however worry me that the correlation between this ratio and TEX86 is quite strong for this core. The authors use the positive correlation to argue that the TEX86 variability is NOT controlled by subsurface-sourced GDGTs, as more subsurface GDGTs should result in colder temperature estimates hence also a negative correlation between TEX86 and GDGT 2/3 ratio. But alternatively this positive correlation might simply mean that the ratio is indeed reflecting temperature change instead of changes in the surface vs subsurface source of GDGT. Indeed, from Figure 7 of Taylor et al. (2013, GPC), it does look like the GDGT 2/3 ratio is not only correlated to water depth but also to SST. If true, this would mean that the authors lose one of the strongest lines of evidence for the subsurface origin of their TEX86 record. I encourage the authors to discuss this possibility in detail.*

REPLY: Taylor et al. (2013) discussed this issue in their paragraph 5.1. They conclude that water depth is the dominant control of GDGT[2/3] ratios in sediments, both in the calibration data set and in the paleo-domain. They also conclude that the positive correlation between SST and GDGT[2/3] (notably the opposite relation to that expected based on homeoviscous adaptation) is an oceanographical artifact. More recent work, as described in lines 179-181, has confirmed this interpretation based on water column profiles. This work finds no evidence of GDGT[2/3] values above 4 in surface waters, making it highly unlikely that that the ratio values we find at Site 959 represent solely surface-derived GDGT assemblages. Accordingly, they must integrate a larger depth range. To comply with the comment of the reviewer, we will include the conclusions that Taylor et al. present in their paragraph 5.1 in section 4.1.1.

*Second, what process would be responsible for the export of GDGTs from subsurface ocean at 350 m or within the AAIW to the seafloor? I appreciate the fact that this is not a calibration study, but the lack of vehicle to export subsurface ocean GDGTs to depths is one of the main criticisms for the HL16 calibration, so I think the reader will be a lot more convinced if the authors can propose some possible mechanisms to explain/demonstrate that a subsurface export of GDGTs is indeed possible.*

REPLY: This is a valid point and has been extensively discussed in previous papers, including Taylor et al. (2013) and Ho and Laepple (2017). Because it represents an issue that is outside the scope of this paper but still important, we will include a few sentences in the introduction, where the issue of deep-production is discussed.

*(3) Is the temporal resolution of the records sufficiently high to assess a 5-10 ky lead/lag relationship between records?*

*The paleoclimate interpretation is largely based on the temporal patterns, so the choice of calibration does not matter much. But what matters here instead is the temporal resolution. Most of the discussion in the last sub-section of Discussion is based on the 5-10 ky lead of TEX86 over benthic d18O, and the lead/lag relationship between pCO2 and temperature records. The lead of TEX86 over benthic d18O is, to my eyes, based on one or two data points. Having said that, I don't think that one can make such a claim given the low-resolution of the proxy records and uncertainty in proxy measurements. I urge the authors to give this some more thought and provide a more balanced discussion taking into account the limitations of their dataset.*

REPLY: this is a valid point and therefore we clearly indicate that this is somewhat speculative in this section, acknowledging that data resolution is the limiting factor (283-284). As suggested below, we will include the inflection points in Figure 5 by plotting lines and will carefully reword the section to convey that we do not consider this data to represent full proof of our hypothesis.

*Specific comments*

*Line 13-15: I would flip the arguments around. First the ecological evidence that Thaumarchaeota / GDGTs occur in the subsurface ocean, then only the circumstantial evidence of the good correlation of TEX86 to temperature from various depths.*

REPLY: We will reconsider wording here to optimize the argument.

*Line 17: "proved" is a bit strong. In my opinion, structural similarities in downcore records are at best circumstantial evidence, not direct proof. What about "can be best assessed in downcore studies"?*

REPLY: we will replace "proved" by "determined"

*Line 166-174: The authors argue that TEX86 records subsurface ocean temperature variability because the long-term trend in TEX86 differs from that of other SST proxies, including UK'37 and Mg/Ca. As the authors noted, these two proxies have their own issues: UK'37 is close to saturation and thus may be insensitive to temperature change, whereas Mg/Ca is susceptible to secular change in seawater Mg/Ca, carbonate chemistry and salinity. I note that the Mg/Ca records are from a sister paper (cp-2021-68), and likely the authors have discussed all the issues in that paper. It would still make life easier for the reader if the authors can briefly summarize these Mg/Ca issues here and why the trend is robust despite these potential caveats. As for UK'37, it might be helpful to also test the Bayspline or other tropical UK'37 calibration (e.g. Sonzogni et al. 1997 DSR II) that has a steeper slope than the one in Prahl and Wakeham (1987 Nature) or Müller et al. (1998 GCA).*

REPLY: we will include a brief summary on the Mg/Ca. We are hesitant to include other tropical UK'37 calibrations. We agree with the criticism of Herbert et al. (2020) regarding the application of UK37 using the calibration suggested: "We hesitate to adopt the Tierney and Tingley (2018) approach because it potentially amplifies noise (difficulty in accurately determining the very small C37:3 peak area at high Uk'37) into signal". We will include this at the end of section 2.2 (methods).

*Line 189-190: Unclear reasoning, please rephrase. Also see my general comment (2).*

REPLY: this was indeed somewhat clumsily phrased. This will be corrected in the revised version.

*Line 206-207: It is difficult to see this in Figure 5. It might be helpful if the authors can illustrate the lead in the figure. Also see my general comment (3). Line 223–224: Again, this is not immediately clear from the figure. Also, how is it established? How is the onset defined? See comment on Line 206-207.*

REPLY: To clarify, we will include lines to indicate the interpreted age of changes discussed.

*Line 240-241: I would rephrase this. Strictly speaking, HL16 calibration does not assume/target any water depths. Instead, they search for calibrations that can reconcile the variability of UK'37 and TEX86. The depth distribution indicates that the most probable depth range is 100-350 m. Thus, instead of using the entire calibration ensemble that includes all depth ranges down to 950 m, one may very well choose one that is calibrated to 100-350 m.*

REPLY: This comment indicates to us that we have to further clarify how we determined the depth distribution of the calibration weights from the calibration ensemble of Ho and Laepple. The calibrations were binned based on depth ranges. In the revised version, we will include a new section in the supplementary information that will clarify our approach, including the depth ranges used.

*Line 242-244: Thaumarchaeotal cell counts and GDGT concentrations vary a lot in space. Are studies from the South Atlantic and Arabian Sea the suitable choice of reference here? If no water column studies are available in the study area, at least tell the reader why it is reasonable to assume that what happens in the water column elsewhere might be applicable to eastern equatorial Atlantic.*

REPLY: We are slightly confused regarding this comment as this sentence summarizes inferences from the globally available data sets of the vertical distribution of Thaumarcheotal cells counts and GDGT concentrations, extensively discussed in the literature. This is used as a model for Site 959, as is the TEX86 calibration from a global set of surface sediment data. To address the concerns of the reviewer, we will reference the Atlantic Ocean study of Sintes et al. (2016, https://doi.org/10.3389/fmicb.2016.00077).

*Line 251-257: Given the hydrography at the study site which is under the influence of episodic monsoon-induced upwelling, I would expect the temperature variability here to be larger than that at pelagic sites like the warm pool. So I think it unlikely that the 1:1 relationship between the surface and subsurface ocean would hold over millions of years, nor is it reasonable to assume that the subsurface temperature variability here would be comparable to global SST change in the tropics.*

REPLY: Here, we merely assess the implications of assuming the 1:1 relationship (tested by Ho & Laepple (their figure S5) but indeed not over million-year time scales). This indicates a SST drop that is consistent with estimates of the global average cooling (so indeed slightly larger than elsewhere in the tropics as the reviewer suggests). We indicate quite clearly that this assumption needs further testing (final 5 lines of this paragraph).

*Line 277-285: See general comment (3).*

REPLY: See our reply to general comment 3

---

## Author Response (AR1)

Dear Dr. Winguth,

We thank you and both reviewers for their comments and suggestions on our paper. We are very happy with their positive and constructive evaluations. Below, we discuss their comments point-by-point and indicate how we have adapted the manuscript.

Sincerely, also on behalf of my co-authors,

Carolien van de Weijst

*Summary*

*The study by van der Weijst et al presents multiproxy-based temperature and productivity records spanning the past 15 Ma. Data are generated from an ODP core recovered from the eastern equatorial Atlantic, at a site that is under the influence of monsoon-induced upwelling. The proxies used are alkenone-based UK'37, archaeal GDGT-based TEX86 and dinocysts. Based on several lines of evidence, the authors argue that the TEX86 proxy records a mixture of surface and subsurface signal, likely that of Antarctic Intermediate Water (AAIW) as also suggested by two previous studies. Assuming that UK'37 reflects SST variability, benthic d18O records NADW variability and TEX86 reflects AAIW variability, in combination with other published temperature records and a pCO$_2$ record, the authors discuss possible mechanisms that lead to M2 glaciation.*

*General comments*

*Overall I find the manuscript very clear and well-written. The topic also fits well within the remit of the journal, thus will likely be of interest to the broad readership of Climate of the Past. Although generally accessible to the reader, I think some arguments can be further improved and/or need further clarification. I hope the authors will find my comments and suggestions helpful in revising the manuscript. Altogether it should amount to moderate revision. Below I outline my major concerns.*

REPLY: We thank reviewer #1 for their compliments regarding our work.

*(1) TEX86 reflects AAIW variability*

*The authors provided several lines of evidence to support their claim that TEX86 reflects subsurface temperature: high GDGT 2/3 ratio, TEX86H SSTs that are out of the modern range and unrealistically large magnitude of change over the last 15 Ma, temporal trends that are more similar to those of benthic d18O than UK'37 and Mg/Ca based on both mixed layer- and thermocline-dwelling species. The discussion is generally convincing.*

*What is however unclear to me is how the authors then link the TEX86 signal to AAIW variability. I find the explanation provided at Line 226-227 to be rather vague. The core-top value of their SubT TEX86 record is ~14ºC, which is a lot warmer than the temperature in AAIW (~5ºC according to Figure 7a) and is in fact closer to that of South Atlantic Central Water. Further, the core of depth range integrated by the HL16 calibration is 100-350 m, which is much shallower than the average depth of AAIW. As the climate interpretation hinges on this claim, the authors need to provide clearer arguments to support their statement. I wonder if it is possible to attribute the relative influence of SACW vs. AAIW using the depth distribution in the calibration used?*

REPLY: We agree with the reviewer that our signal represents SACW. In other words, our TEX86 signal tracks Southern Ocean surface temperature variability imprinted in SACW. So to comply with the comment of the reviewer, we have changed reference of AAIW in the remainder of the manuscript to SACW, thereby noting that both represent a Southern Hemisphere, higher latitude climate signal. We have adapted this throughout the manuscript.

*(2) Subsurface export of GDGTs*

*Most sedimentary GDGT 2/3 ratios along the core are > 5. The authors argue that this indicates that GDGTs in the sediment core are partially sourced from the subsurface ocean. I acknowledge that the GDGT 2/3 ratio is a routinely used indicator to cull TEX86 data, it does however worry me that the correlation between this ratio and TEX86 is quite strong for this core. The authors use the positive correlation to argue that the TEX86 variability is NOT controlled by subsurface-sourced GDGTs, as more subsurface GDGTs should result in colder temperature estimates hence also a negative correlation between TEX86 and GDGT 2/3 ratio. But alternatively this positive correlation might simply mean that the ratio is indeed reflecting temperature change instead of changes in the surface vs subsurface source of GDGT. Indeed, from Figure 7 of Taylor et al. (2013, GPC), it does look like the GDGT 2/3 ratio is not only correlated to water depth but also to SST. If true, this would mean that the authors lose one of the strongest lines of evidence for the subsurface origin of their TEX86 record. I encourage the authors to discuss this possibility in detail.*

REPLY: Taylor et al. (2013) discussed this issue extensively in their paragraph 5.1. They conclude that water depth is the dominant control of GDGT[2/3] ratios in sediments, both in the calibration data set and in the paleo-domain. They also conclude that the positive correlation between SST and GDGT[2/3] (notably the opposite relation to that expected based on homeoviscous adaptation) is an oceanographical artifact. More recent work, as described in lines 179-181, has confirmed this interpretation based on water column profiles. This work finds no evidence of GDGT[2/3] values above 4 in surface waters, making it highly unlikely that that the ratio values we find at Site 959 represent solely surface-derived GDGT assemblages. Accordingly, they must integrate a larger depth range. To comply with the comment of the reviewer, we have included these insights in section 4.1.1.

*Second, what process would be responsible for the export of GDGTs from subsurface ocean at 350 m or within the AAIW to the seafloor? I appreciate the fact that this is not a calibration study, but the lack of vehicle to export subsurface ocean GDGTs to depths is one of the main criticisms for the HL16 calibration, so I think the reader will be a lot more convinced if the authors can propose some possible mechanisms to explain/demonstrate that a subsurface export of GDGTs is indeed possible.*

REPLY: This is a valid point and has been extensively discussed in previous papers, including Taylor et al. (2013) and Ho and Laepple (2017). However, it is well-established that export of organic matter takes place below the mixed layer, but decreases exponentially below the photic zone (e.g., Martin et al., 1987; Middelburg, 2020). We have included the following section in the introduction, where the issue of deep-production and export is discussed, which are followed by the already included indicators based on which contributions from deeper water layers has been shown:

*Although GDGTs may be dominantly produced at depths below the mixed layer, the efficiency of organic carbon export towards the sediment decreases exponentially with depth below the photic zone as shown with sediment trap data and simulations (e.g., Martin et al., 1987; Middelburg et al., 2020), which for GDGTs is supported by in situ measurements (e.g., Wüchter et al., 2005). However, as the production of GDGTs in the mixed layer is generally low relative to production within and below the thermocline, GDGTs produced below the mixed layer might still comprise a significant component of the sedimentary assemblage (e.g., Taylor et al., 2013; Ho and Laepple, 2017).*

***(3) Is the temporal resolution of the records sufficiently high to assess a 5-10 ky lead/lag relationship between records?***

*The paleoclimate interpretation is largely based on the temporal patterns, so the choice of calibration does not matter much. But what matters here instead is the temporal resolution. Most of the discussion in the last sub-section of Discussion is based on the 5-10 ky lead of TEX86 over benthic d18O, and the lead/lag relationship between pCO2 and temperature records. The lead of TEX86 over benthic d18O is, to my eyes, based on one or two data points. Having said that, I don't think that one can make such a claim given the low-resolution of the proxy records and uncertainty in proxy measurements. I urge the authors to give this some more thought and provide a more balanced discussion taking into account the limitations of their dataset.*

REPLY: this is a valid point and therefore we already clearly indicated that this is somewhat speculative in this section, acknowledging that data resolution is the limiting factor (283-284). As suggested below, we will include

the inflection points in Figure 5 by plotting lines. We have included the following sentence at the end of the section to clarify: *Higher resolution records and correlations would be required to fully test this hypothesis.*

***Specific comments***

*Line 13-15: I would flip the arguments around. First the ecological evidence that Thaumarchaeota / GDGTs occur in the subsurface ocean, then only the circumstantial evidence of the good correlation of TEX86 to temperature from various depths.*

REPLY: Done.

*Line 17: "proved" is a bit strong. In my opinion, structural similarities in downcore records are at best circumstantial evidence, not direct proof. What about "can be best assessed in downcore studies"?*

REPLY: we have now used 'assessed' in the rewording of the section.

*Line 166-174: The authors argue that TEX86 records subsurface ocean temperature variability because the long-term trend in TEX86 differs from that of other SST proxies, including UK'37 and Mg/Ca. As the authors noted, these two proxies have their own issues: UK'37 is close to saturation and thus may be insensitive to temperature change, whereas Mg/Ca is susceptible to secular change in seawater Mg/Ca, carbonate chemistry and salinity. I note that the Mg/Ca records are from a sister paper (cp-2021-68), and likely the authors have discussed all the issues in that paper. It would still make life easier for the reader if the authors can briefly summarize these Mg/Ca issues here and why the trend is robust despite these potential caveats. As for UK'37, it might be helpful to also test the Bayspline or other tropical UK'37 calibration (e.g. Sonzogni et al. 1997 DSR II) that has a steeper slope than the one in Prahl and Wakeham (1987 Nature) or Müller et al. (1998 GCA).*

REPLY: we have included a clarifying sentence on the Mg/Ca record, covering data quality and calibration. We are hesitant to include other tropical UK'37 calibrations. We agree with the criticism of Herbert et al. (2020) regarding the application of UK37 using the calibration suggested: "We hesitate to adopt the Tierney and Tingley (2018) approach because it potentially amplifies noise (difficulty in accurately determining the very small C37:3 peak area at high Uk'37) into signal". We have included this at the end of section 2.2 (methods).

*Line 189-190: Unclear reasoning, please rephrase. Also see my general comment (2).*

REPLY: this was indeed somewhat clumsily phrased. This has been rewritten as follows: *The ratio of GDGT[2] to GDGT[3] in samples does not directly affect the TEX$_{86}$-index because they are both included in the denominator and divisor (Equation 1). If the abundance of GDGT[2] increases relative to that of GDGT[3] with export depth, then samples with a higher GDGT[2/3] ratio should be associated with lower TEX$_{86}$ values.*

*Line 206-207: It is difficult to see this in Figure 5. It might be helpful if the authors can illustrate the lead in the figure. Also see my general comment (3). Line 223–224: Again, this is not immediately clear from the figure. Also, how is it established? How is the onset defined? See comment on Line 206-207.*

REPLY: To clarify, we have included lines to indicate the interpreted age of changes discussed.

*Line 240-241: I would rephrase this. Strictly speaking, HL16 calibration does not assume/target any water depths. Instead, they search for calibrations that can reconcile the variability of UK'37 and TEX86. The depth distribution indicates that the most probable depth range is 100-350 m. Thus, instead of using the entire calibration ensemble that includes all depth ranges down to 950 m, one may very well choose one that is calibrated to 100-350 m.*

REPLY: In the revised version, we have included a new section in the supplementary information that will clarify our approach, including the depth ranges used.

*Line 242-244: Thaumarchaeotal cell counts and GDGT concentrations vary a lot in space. Are studies from the South Atlantic and Arabian Sea the suitable choice of reference here? If no water column studies are available in*

*the study area, at least tell the reader why it is reasonable to assume that what happens in the water column elsewhere might be applicable to eastern equatorial Atlantic.*

REPLY: We are slightly confused regarding this comment as this sentence summarizes inferences from the globally available data sets of the vertical distribution of Thaumarcheotal cells counts and GDGT concentrations, extensively discussed in the literature. This is used as a model for Site 959, as is the TEX86 calibration from a global set of surface sediment data. To address the concerns of the reviewer, we have now also cited the Atlantic Ocean study of Sintes et al. (2016, https://doi.org/10.3389/fmicb.2016.00077).

*Line 251-257: Given the hydrography at the study site which is under the influence of episodic monsoon-induced upwelling, I would expect the temperature variability here to be larger than that at pelagic sites like the warm pool. So I think it unlikely that the 1:1 relationship between the surface and subsurface ocean would hold over millions of years, nor is it reasonable to assume that the subsurface temperature variability here would be comparable to global SST change in the tropics.*

REPLY: Here, we merely assess the implications of assuming the 1:1 relationship (tested by Ho & Laepple (their figure S5) but indeed not over million-year time scales). This indicates a SST drop that is consistent with estimates of the global average cooling (so indeed slightly larger than elsewhere in the tropics as the reviewer suggests). We indicate quite clearly that this assumption needs further testing (final 5 lines of this paragraph). To alleviate the concerns of the reviewers, we have included the specific assumptions on which this approach hinges in the beginning of this paragraph: The ratio between temperature change in the surface and subsurface ocean is 1:1 when averaged across many sites and on longer timescales (Ho and Laepple, 2016). Under the assumption that the (integrated) depth of GDGT sourcing and the structure of the water column was stable, a depth-integrated $TEX_{86}$ record might still be of value to assess SST variability.

*Line 277-285: See general comment (3).*

REPLY: See our reply to general comment 3

Dear Editor,

We thank reviewer #2 for their comments and suggestions on our paper. Below, we discuss these point-by-point and indicate how we intend to adapt the manuscript in a revised version.

Sincerely, also on behalf of my co-authors,

Carolien van de Weijst

*Summary:*

*The manuscript presents a new 15 Myr TEX86 record from ODP Site 959 in the Gulf of Guinea. Comparing the TEX86 record to other proxies of the same core and other sediment cores, it investigates the source of the recorded TEX86 variability and concludes that for this site, TEX86 is mainly a subsurface temperature proxy. Applying a subsurface calibration, it then discusses the climatic implications of the record and suggests that the M2 glacial was marked by AAIW cooling during an austral summer insolation minimum, and that decreasing CO 2 levels were a feedback, not the initiator, of glacial expansion.*

*The manuscript is well written and fits in the scope of Climate of the past. The paper is twofold; Based on the multiproxy comparison / interpretation, it provides an attribution of the source for the Tex86 signal for the specific core and supports a rather controversial hypothesis; that the Tex86 signal could originate from subsurface temperature variations and would thus also have a different temperature sensitivity than assumed in most studies. In addition to this proxy attribution part, it also provides a paleoclimatic interpretation of the record which is largely independent on the calibration but hinges on the timing of the records.*

*I have only three comments and would recommend the manuscript for publication in CP*

REPLY: We thank the reviewer for their positive evaluation and thoughtful comments.

• *The manuscript concludes that the Tex86 record for this site should be interpreted as subsurface temperature (SubST) and provides a range of convincing arguments supporting this interpretation (magnitude of variations, similarity to benthic records, magnitude of trend, high GDGT 2/3 ratio). Despite this, in large parts, the manuscript clings on the SST interpretation and discusses Tex86 SST and shows Tex86 SST (Line 137, Line 164, Figure 2,3,4,5). At least, to me this is confusing and inconsistent as either the authors interpret the record as subsurface temperature and the SST interpretation is only used as a first guess before the depth attribution; or the authors interpret the record as SST… which would than change large parts of the manuscript and the conclusions. I would thus suggest clarifying that the first result is the Tex86H record (without calibration) and the calibration itself is an interpretation. As an example, in Line 135, instead of 'Between 15 and 11 Ma, TEX86H-SST fluctuates…", one could write "Interpreting TEX86H as SST proxy, the inferred SSTs (here called TEX86H-SST) fluctuate. For the figures, the authors should check if the SST interpretation is needed in all of the figures.*

REPLY: This is a good point and it was one of the points of internal discussion as we were drafting the manuscript. The intention here was to clarify that the assumption that TEX86 reflects SST leads to inconsistencies with all lines of evidence, to subsequently conclude that it cannot hold and subsurface calibrations are required. We have now specifically laid out our approach in the methods section and the start of the results section that we, by convention, first explore our TEX86 results following an SST calibration (TEX86H) and include subsurface calibrations as soon as we have established an integrated surface to subsurface sourcing for the GDGTs. This should avoid confusion for the reader.

• *In several occasions, (e.g. L25, L264, L315) the authors argue that the "depth-integrated TEX86 record can potentially be used to infer SST variability, because subsurface temperature variability is generally tightly linked to SST variability". However, they also find that the time-series of the Tex86 record differs from the surface/thermocline temperature derived from other proxies and argue that this is due to the depth-integration of their record. Both statements seem contradictory. As support, the authors (L249) cite Ho and Laepple, 2016:*

*"The ratio between temperature change in the surface and subsurface ocean is 1:1 when averaged across many sites and on longer timescales". However, as the authors show themselves in their presented multiproxy records, this does not apply to any single site or on details as the phasing of temperature variations. It is also unclear why such a translation to SST would be needed as a subsurface proxy also provides important information on the climate dynamics; can be used to validate models etc. Thus, I would suggest removing the parts on the possible translation of SubST to SST or to provide arguments why the authors think such a conversion is possible and useful as the conversion is clearly not just an offset.*

REPLY: Also this is a good point. Essentially, we aim to present this as an option for assessing SST based on TEX86, We consider it a valuable avenue to explore for the community and therefore prefer to retain it. We have, however, further clarified the set of assumptions that underly the approach in the first part of this section.

• *One main argument for the choice of the calibration is the amplitude of the reconstructed variations. This is mainly demonstrated in Figure S1. As the choice of the calibration is a major result of the manuscript, I would suggest moving this Figure into the main manuscript.*

REPLY: This has been done